# Identification of *hsa_circ_0018905* as a New Potential Biomarker for Multiple Sclerosis

**DOI:** 10.3390/cells13191668

**Published:** 2024-10-09

**Authors:** Valeria Lodde, Ignazio Roberto Zarbo, Gabriele Farina, Aurora Masia, Paolo Solla, Ilaria Campesi, Giuseppe Delogu, Maria Rosaria Muroni, Dimitrios Tsitsipatis, Myriam Gorospe, Matteo Floris, Maria Laura Idda

**Affiliations:** 1Department of Biomedical Sciences, University of Sassari, Sassari 07100, Italy; vlodde@uniss.it (V.L.); icampesi@uniss.it (I.C.); giuseppe@uniss.it (G.D.); mfloris@uniss.it (M.F.); 2Department of Medicine, Surgery and Pharmacy, University of Sassari, Sassari 07100, Italy; irzarbo@uniss.it (I.R.Z.); masia.aurora@tiscali.it (A.M.); psolla@uniss.it (P.S.); mrmuroni@uniss.it (M.R.M.); 3Unit of Clinical Neurology, AOU, Sassari 07100, Italy; gabriele.farina@aouss.it; 4Department of Medical Sciences and Public Health, University of Cagliari, Cagliari 09042, Italy; 5Laboratory of Genetics and Genomics, National Institute on Aging Intramural Research Program, National Institutes of Health, Baltimore, MD 21224, USA; dimitrios.tsitsipatis@nih.gov (D.T.); gorospem@grc.nia.nih.gov (M.G.)

**Keywords:** circular RNAs, multiple sclerosis, biomarkers, new diagnosis, immune regulation

## Abstract

Multiple sclerosis (MS) is a demyelinating autoimmune disease characterized by early onset, for which the interaction of genetic and environmental factors is crucial. Dysregulation of the immune system as well as myelinization-de-myelinization has been shown to correlate with changes in RNA, including non-coding RNAs. Recently, circular RNAs (circRNAs) have emerged as a key player in the complex network of gene dysregulation associated with MS. Despite several efforts, the mechanisms driving circRNA regulation and dysregulation in MS still need to be properly elucidated. Here, we explore the panorama of circRNA expression in PBMCs purified from five newly diagnosed MS patients and five healthy controls (HCs) using the Arraystar Human circRNAs microarray. Experimental validation was then carried out in a validation cohort, and a possible correlation with disease severity was tested. We identified 64 differentially expressed circRNAs, 53 of which were downregulated in PBMCs purified from MS compared to the HCs. The discovery dataset was subsequently validated using qRT-PCR with an independent cohort of 20 RRMS patients and 20 HCs. We validated seven circRNAs differentially expressed in the RRMS group versus the HC group. *hsa_circ_0000518*, *hsa_circ_0000517*, *hsa_circ_0000514*, and *hsa_circ_0000511* were significantly upregulated in the MS group, while *hsa_circ_0018905*, *hsa_circ_0048764*, and *hsa_circ_0003445* were significantly downregulated; Among them, the expression level of *hsa_circ_0018905* was significantly decreased in patients showing a higher level of disability and in progressive forms of MS. We described the circRNAs expression profile of PBMCs in newly diagnosed MS patients and proposed *hsa_circ_0018905* as potential MS biomarker.

## 1. Introduction

Multiple sclerosis (MS) is an early-onset complex demyelinating autoimmune disease of the central nervous system (CNS) characterized by chronic inflammation, neuronal loss, and axonal damage [1]. Clinically, MS has been classified into four main categories based on disease progression: clinically isolated syndrome (CIS); relapsing–remitting MS (RRMS), comprising 80% of patients; primary progressive MS (PPMS); and secondary progressive MS (SPMS) [2].

Recent data indicate that the frequency of MS follows a general north–south gradient, with the prevalence of MS tending to be higher further away from the equator. Exceptions include specific populations, such as the Sardinians, characterized by having among the highest MS prevalence in the world despite its geographical localization [3]. Indeed, the prevalence of MS in Sardinia is about 361/100,000 inhabitants and the peculiar genetic background of Sardinians could partly explain this record [4].

Furthermore, it is widely accepted by the scientific community that genetic background alone is not enough to trigger disease onset, and that genetic, environmental, and epigenetic factors interact to drive MS pathogenesis [5,6]. The results from genome-wide association studies (GWAS) have identified HLA-DRB1*15:01 as one of the most prominent factors influencing MS onset; in addition, more than 236 HLA-independent genetic variants have been associated with an increased risk of MS. Most of the identified variants alter regulatory non-coding regions and influence gene expression. In general, genomic studies suggested a plethora of loci and genes with small effects on autoimmunity risk, mainly implicated in immunological (both innate and adaptive) and neurological pathways [7,8]. Environmental factors, such as vitamin D deficiency, Epstein–Barr virus (EBV), smoking, and obesity, have also been linked to MS onset [4].

Despite the fact that the underlying causes of MS are still not well understood, recently a relevant role of non-coding RNA (ncRNA) and RNA processing was identified as a key component of the molecular mechanisms potentially involved in MS pathogenesis [9]. Additionally, due to their molecular features, ncRNAs have also been suggested as new biomarkers for MS [10]. Circular RNAs (circRNAs) are a class of ncRNAs that have a unique closed-loop structure, making them more resistant than linear RNAs to degradation by ribonucleases (RNases) [11]. Notably, the type of circRNAs and their abundance vary depending on the developmental stage, tissue, age, and disease stage. According to their biogenesis, circRNAs can be categorized as exonic circRNAs (ecircRNAs), circular intronic circRNAs (ciRNAs), and exonic-intronic circRNAs (EIciRNAs) [11]. It is now widely acknowledged that circRNAs play crucial roles in cell proliferation and apoptosis, as well as in cell cycle regulation and regulation of immune cells and cytokines production [12]. Of note, circRNAs are predominantly found in the cytoplasm, but many are also localized and enriched in synapses and play significant roles in synaptic plasticity and neuronal functions [13]. Finally, circRNAs can influence the pathophysiology of several diseases through different functions, including acting as binding substrates for microRNAs (miRNAs) and RNA-binding proteins (RBPs), as well as by functioning as regulators of transcription, mRNA metabolism, and translation [14].

Accumulating evidence reveals that circRNAs elicit prominent roles in pathologies and play a key role at the interface of the immune system and CNS. One of the first identified dysregulated circRNA in MS patients was *hsa_circ_0106803*, which originates from the GSDMB gene. *Hsa_circ_0106803* was shown to be upregulated in peripheral blood mononuclear cells (PBMCs) of RRMS patients as compared to healthy controls (HCs) [15]. Later, Iparraguirre and colleagues detected 406 differentially expressed circRNAs through a microarray analysis on leucocytes of four RRMS patients and four HCs; *hsa_circ_0005402* and *hsa_circ_0035560*, two downregulated circRNAs derived from the *ANXA2* gene, were then validated [16]. Next, the enrichment of circRNAs in the MS transcriptome was studied and a circRNA derived from the MS-associated *STAT3* gene was identified [9]. In addition to genetic variants, mechanisms influencing circRNA expression levels, including epigenetic factors and regulatory networks involving other linear lncRNAs, have been proposed [17,18].

Importantly, the levels of expressed circRNAs can be influenced by disease progression and associated therapies. Therefore, we decided to study circRNAs differentially expressed in PBMCs from five newly diagnosed MS patients and five HCs. Using the Arraystar Human Circular RNA Array, we identified 64 differentially expressed circRNAs (11 upregulated and 53 downregulated) in MS compared to the HCs. Evidence gained from experimental validation revealed that *hsa_circ_0018905* may be used as a new blood biomarker for MS.

## 2. Materials and Methods

### 2.1. Study Design

Whole-blood samples were collected from a total of 54 subjects enrolled in the study: 34 patients diagnosed with MS, and 20 HCs, consisting of age- and sex-matched individuals. The study was designed and organized to have three different cohorts: (i) a discovery cohort comprised of 5 newly diagnosed MS patients, who had no comorbidities and did not receive any pharmacological treatment, and 5 HCs matched for age and sex; (ii) a validation cohort, consisting of 20 adults diagnosed with MS with no comorbidities and who did not receive any treatment, and 20 HCs matched for age and sex; and (iii) a third cohort of MS patients with mild and severe disease, consisting of 8 RRMS patients newly diagnosed with Expanded Disability Status Scale (EDSS) < 4.5, 6 RRMS patients with EDSS > 4.5, and 8 SPMS patients with EDSS > 4.5 (Appendix A). All MS patients were recruited from the University Hospital of Sassari (Italy). Total RNA, derived from PBMCs, was isolated as described below and used for the initial microarray analysis (discovery cohort) and validation steps ((ii) and (iii)) using reverse transcription (RT) followed by quantitative (q) PCR.

### 2.2. RNA Extraction

PBMCs were purified using 15 mL of blood collected using Vacutainer CPT tubes (BD) from each donor following the manufacturer’s instructions. Total RNA from PBMCs was then isolated using the RNeasy Mini Kit (Qiagen, Hilden, Germany). Human serum was obtained from patients and volunteers and processed as follows. Briefly, vacutainers containing clot activator were rested upright for 60 min to allow RBCs to clot. The RBC clot was subsequently pelleted by centrifugation at 1000× *g* for 10 min and serum was collected. The collected supernatant was then centrifuged at 5000× *g* for 10 min to pellet cell fragments and other debris. Serum samples were pooled and stored in 200 µL aliquots at −80 °C prior to analysis. RNA from serum was subsequently isolated using miRNeasy Serum/Plasma Advanced kit (Qiagen, Hilden, Germany) following the manufacturer’s guidelines.

### 2.3. RNA Digestion, Amplification, Labeling, and Hybridization

Total RNA extracted from the discovery cohort was used for microarray analysis. Sample labeling and array hybridization were performed according to the manufacturer’s protocol (Arraystar Inc., Rockville, MD, USA). Briefly, total RNA was digested with RNase R (Epicentre, Inc., Madison, WI, USA) to remove linear RNAs and enrich in circular RNAs. Then, the enriched circular RNA pool was transcribed into fluorescent cRNA utilizing the random priming method, and then amplified (Arraystar Super RNA Labeling Kit; Arraystar). The labeled cRNAs were purified using the RNeasy Mini Kit (Qiagen, Hilden, Germany). The concentration and specific activity of the labeled cRNAs (pmol Cy3/μg cRNA) were measured by NanoDrop ND-1000; 1 μg of each labeled cRNA was fragmented by adding 5 μL Blocking Agent and 1 μL of Fragmentation Buffer, then heated the mixture at 60 °C for 30 min. Finally, 25 μL of Hybridization buffer was added to dilute the labeled cRNA, and 50 μL of hybridization solution was dispensed into the gasket slide and assembled to the circRNA expression microarray slide. The slides were incubated for 17 h at 65 °C in an Agilent Hybridization Oven. The hybridized arrays were washed, fixed, and scanned using the Agilent Scanner G2505C (Agilent Technologies, Santa Clara, CA, USA).

### 2.4. Reverse Transcription (RT)-Quantitative (q)PCR Analysis

First-strand cDNA synthesis was performed using Maxima reverse transcriptase (Thermo Fisher, Waltham, MA, USA) and random hexamers. The generated cDNA was diluted tenfold and used as a template for RT-qPCR analysis using SYBR Green mix (Kapa Biosystems, Wilmington, MA, USA). Relative levels of RNA were calculated using the 2^−ΔΔCt^ method and the levels of *GAPDH mRNA* were used for normalization. Results are reported using the fold induction value (FI) as compared to the control (CTR). Gene-specific primer pairs are listed in Table 1.

### 2.5. Prediction of circRNA-miRNA and circRNA-RBP Interactions

The spliced sequences of the validated circRNAs were provided as input in the prediction tool miRanda software (v3.3a) to identify circRNA-associated miRNAs [19]. The RBPs potentially binding to circRNAs were predicted by the Circular RNA Interactome (CircInteractome, https://circinteractome.nia.nih.gov/, accessed on 15 January 2024) based on CLIP data sets [20].

### 2.6. CircRNA-microRNA-mRNA Network

Bioinformatic analyses were performed to identify potential miRNA targets for each differentially expressed circRNA using miRanda software v3.3a [19]. We also speculated on the biological functions of each circRNA by prediction analysis of proteins potentially affected by putative regulatory networks of circRNA-miRNA-mRNA. The interactions of miRNAs and mRNAs were identified using mirTarBase software (release 9.0 beta) [21]. Finally, we selected the interaction of miRNAs and MS-associated genes described in GWAS analysis from the International Multiple Sclerosis Genetics Consortium [6]. Cytoscape v3.9.0 [22] was then utilized to visualize circRNA-miRNA-mRNA networks.

### 2.7. Statistical Analysis

Hybridized arrays were scanned, and images were imported into Agilent Feature Extraction software (version 11.0.1.1) for raw data extraction. Quantile normalization of raw data and subsequent data processing were performed using the R software (version 4.4.1) package (Arraystar). After quantile normalization of the raw data, low-intensity filtering was performed, and circRNAs that had flags in at least 1 out of 12 samples in “P” or “M” (“All Targets Value”) were retained for further analyses. CircRNAs with absolute fold changes ≥ 1.5 and *p*-value ≤ 0.05 between the MS and HC samples were selected as significantly differentially expressed. Hierarchical clustering analysis based on expression levels of differentially expressed circRNAs was performed using Java Treeview (Stanford University School of Medicine, Stanford, CA, USA). In the scatter plot depicting the circRNA expression, the horizontal lines represent the medians.

### 2.8. Statistics and Data Representation

Roc curves have been generated using the pROC package available in the R software (version 4.4.1). Data were presented as the means ± standard deviations, and all experimental data were analyzed using GraphPad Prism 9 (GraphPad Software, La Jolla, CA, USA). Differences in RNA levels among groups were evaluated using a two-tailed Student’s *t*-test, and *p* ≤ 0.05 was considered statistically significant.

## 3. Results

### 3.1. Analysis of circRNAs Expressed in MS Patients Using circRNA Arrays

To explore the levels of expressed circRNAs in healthy individuals and newly diagnosed RRMS patients, we performed a microarray analysis of the circRNAs present in PBMCs using the Arraystar Human Circular RNA Array. We enrolled five RRMS and five HC age- and sex-matched subjects in the discovery cohort (Table 2). To mitigate potential confounding variables in the analysis, all donors were negative for comorbidity and did not receive drug treatment. To avoid effects of geographic origin, all the enrolled individuals were from Sardinia, with at least three grandparents of Sardinian ethnicity. For the validation step, the cohort was extended as described below (Table 2).

The Arraystar circRNAs Array was designed to identify 13,617 circRNAs, which were analyzed using the Agilent Feature Extraction software. Differential expression analysis identified a total of 64 dysregulated circRNAs in MS patients at the time of diagnosis, as compared to the HCs, of which 11 were upregulated and 53 were downregulated (Figure 1A). A complete list of dysregulated circRNAs is available in Appendix A. Hierarchical clustering revealed differentially expressed circRNAs between the MS and HCs; as expected, there was a prevalence of downregulated circRNAs in MS patients (Figure 1A,B). Notably, using the differentially expressed circRNAs, we were able to classify individuals as either MS or HC (Figure 1B).

The expression levels of circRNAs showing the most significant difference between the MS and HCs in the discovery cohort are reported in Figure 1C. *Hsa_circ_0000518* was found to be the most upregulated (FI = 2.3; *p* = 0.00073), while *hsa_circ_0003596* was the most downregulated (FI = 3.4; *p* = 0.006). Most of the upregulated circRNAs originate from the parental gene *RPPH1* which encodes an lncRNA component of the ribonuclease P RNA; interestingly, the lncRNA *RPPH1* has been associated with MS and circRNAs at different levels [16,23] (Figure 1C).

The identified differentially expressed circRNAs were distributed on all chromosomes, with chromosome 1 containing the highest number of circRNAs (seven total) (Appendix A). Some upregulated circRNAs were generated by genes located at chromosome 14, while downregulated circRNAs were generated by genes located at several chromosomes, with chromosomes 1, 11, and 19 containing the highest number (Figure 2A–C). Interestingly, the majority of the identified circRNAs mapped within exons, with the upregulated circRNAs belonging to sense overlapping (64%) and exonic (36%) circRNA classes, while downregulated circRNAs were mainly exonic (87%) (Figure 2B–D).

### 3.2. Validation of circRNA Expression Profiles by RT-qPCR

To validate the circRNA candidates identified by the microarray analysis, we performed RT-qPCR-mediated detection of circRNAs using divergent primers (Table 1) and samples from the validation cohort (Table 2) composed of 12 female and 8 male RRMS patients, and 20 HCs (10 female and 10 male); none of the patients or HC individuals received any drug treatment. The expression trend of the selected circRNAs was consistent with the microarray results: RT-qPCR data revealed that *hsa_circ_0000518* (FI = 1.63; *p* = 0.003), *hsa_circ_0000517* (FI = 3.93; *p* = 6.66 × 10^−7^), *hsa_circ_0000514* (FI = 1.45; *p* = 0.004), and *hsa_circ_0000511* (FI = 1.65; *p* = 0.005) were significantly upregulated in the MS group as compared with the HCs, with *hsa_circ_0000517* showing the highest upregulation. Moreover, *hsa_circ_0018904* (FI = 0.39; *p* = 4.38 × 10^−16^), *hsa_circ_0048764* (FI = 0.51; *p* = 1.82 × 10^−8^), and *hsa_circ_0003445* (FI = 0.61; *p* = 2.84 × 10^−9^) were significantly downregulated in the validation cohort. No significant difference in the expression levels of *hsa_circ_0006853* or *hsa_circ_0011998* was found between the MS and HCs (Figure 3A). Next, we examined the relative abundance of the parental linear transcript for each host gene. In contrast to the upregulation of the cognate circRNAs, the levels of *RPPH1* did not significantly change in samples from MS patients compared to HCs, suggesting post-transcriptional regulation of the four circRNAs analyzed. By contrast, for the downregulated transcripts, we observed a coherent change in expression among circRNAs and corresponding linear transcripts (Figure 3B).

### 3.3. PBMC circRNA Expression Correlation with Serum

PBMCs are present in blood, where they actively release extracellular vesicles (EVs) containing several classes of ncRNAs, including circRNAs. Thus, we decided to check whether the differentially expressed circRNAs can be detected in serum, and whether the levels of the circRNAs detected in PBMCs may be reflected in serum EVs. Therefore, serum was isolated and quickly stored at −80 °C; RNA was then isolated and measured using RT-qPCR analysis. The levels of the circRNAs *hsa_circ_0000518*, *hsa_circ_0000514*, *hsa_circ_0000511*, and *hsa_circ_0006853* increased in serum purified from MS patients compared to HCs, in line with the observation in PBMCs; *hsa_circ_0000517* was not detected in serum. In contrast, the levels in serum of *hsa_circ_0111998*, *hsa_circ_0018904*, *hsa_circ_0048764*, and *hsa_circ_0003445* were not influenced by disease status (Figure 4A) in serum samples. Interestingly, we also analyzed the linear mRNA and observed that all mRNAs analyzed could be detected in serum, and that *LYPLAL1* and *SAMD8* mRNAs are slightly but significantly more abundant in serum isolated from MS patients (Figure 4B).

### 3.4. Identification of the miRNA and RBP Targets and circRNA-miRNA-mRNA Network

Highly abundant circRNAs have been shown to function as ‘sponges’ to sequester miRNAs and RBPs, and in turn regulate gene expression [24]. Accordingly, a miRNA target prediction program, miRanda, was used to identify in silico potential miRNA binding sites within the sequence of the circRNAs; many miRNAs were identified to have predicted miRNA response elements (MREs), and the five top-scoring miRNAs (based on the miRanda “max score”) are listed (Figure 5A). For the upregulated circRNAs, an overlap on the listed miRNAs can be observed due to the common parental gene, *RPPH1*. To identify candidate RBPs interacting with the selected circRNAs, we used CircInteractome, a web tool that was developed to identify RBP-binding sites on circRNA sequences using CLIP-seq datasets from various sources [20]. Several RBPs have been identified to have potential binding sites in circRNAs. On average, circRNAs have seven different binding sites for different RBPs, and among them, the Argonaute family and FUS are the most highly represented and potentially interact with 70% of the identified and validated circRNAs (Figure 5B). The circRNA *hsa_circ_0048764* had the highest number of binding sites for RBPs in accordance with its greater length (1333 bp, with 73 binding sites).

Next, we decided to study the circRNA-miRNA-mRNA network and used the identified miRNAs to search for potential mRNA targets affected by the dysregulation of the identified circRNA. mRNA targets were selected using the list of candidate genes generated by the International Multiple Sclerosis Genetics Consortium [6]. Combining the information described above (circRNA, miRNA, and mRNA) and the mirTarBase interface [21], we constructed a putative circRNA-miRNA-mRNA network underlying dysregulated pathways in MS patients. As presented in Figure 6A, regulatory networks generated from the five validated upregulated circRNAs are predicted to result in altered expression of key proteins associated with the MS pathogenesis, namely CXCR4, FOXP1, and RREB1. Specifically, *hsa_circ_0000518*, *hsa_circ_0000517*, and *hsa_circ_0000511* may bind to different miRNAs that regulate the same gene RREB1 (Ras-responsive element-binding protein 1) which is a risk gene for MS; recent evidence identified a key role of RREB1 as a positive regulator of many genes essential for neuron survival in mammalian brain [25]. The network generated using data from the downregulated circRNAs (Figure 6B) comprised three circRNA-miRNA-mRNA axes predicted to control the abundance of pivotal proteins in MS pathogenesis, such as STAT3, ZFP36L1, and IKF3. This network highlights a key role of the miRNA-4270, which might be sponged by *hsa_circ_0048764*, and may affect the expression of the *SLCO30A7*, *STAT3*, *ZFP36L1*, and *VCAM1* mRNAs. The protein encoded by *ZFP36L1* (zinc finger protein 36-C3H-type-like 1) mRNA participates in mRNA degradation and translational repression. It limits the expression of a number of critical proteins involved in the regulation of immune function [26] and has been associated with several autoimmune diseases [27,28,29]. *hsa_circ_0018905* and *hsa_circ_0003445* bind miRNA-6727-3p and miRNA-612 to potentially affect the expression of IKZF3 (proteins Aiolos and Ikaros), which regulate lymphoid and myeloid cell development as well as immune homeostasis, in particular maturation and differentiation of B cells. Interestingly, IKZF3 was found to be upregulated in PBMCs of patients with MS during the relapse phase [30]. Notably, several circRNAs, including the upregulated *hsa_circ_0000518* and *hsa_circ_0000517*, as well as the downregulated *hsa_circ_0048764* and *hsa_circ_0003445*, may interact with distinct miRNAs that influence the expression of the PLEC gene, an essential gene in maintaining tissue integrity and elasticity in the brain [31]. Whether these predicted regulatory paradigms materialize in MS requires dedicated studies in which the copy numbers of each participating component of the complex (circRNA-miRNA-mRNA) are carefully measured, the physical interactions between the various RNAs are evaluated, and gain- and loss-of-function experiments are carried out to validate the proposed regulatory events.

### 3.5. PBMC circRNA Expression and Correlation with Disease Severity

To assess the potential implication of the identified circRNAs in the pathophysiology of the disease, we decided to evaluate the correlation between the expression of circRNA candidates and disease severity. We thus performed RT-qPCR analysis using a different cohort consisting of eight newly diagnosed RRMS patients with EDSS < 4.5, six RRMS patients with EDSS > 4.5, and eight SPMS patients with EDSS > 4.5 (Table 3).

The expression levels of *hsa_circ_0000511* (FI = 0.73; *p* = 0.004 for RR EDSS > 4.5, FI = 0.62; *p* = 0,005 for SP EDSS > 4.5) and *hsa_circ_0018905* (FI = 0.58; *p* = 0.0004 for RR EDSS > 4.5, FI = 0.70; *p* = 0.01 for SP EDSS > 4.5) were significantly lower in RRMS and SPMS with EDSS > 4.5 as compared with RRMS with EDSS < 4.5, thus showing a decrease in the target circRNAs with the increase in MS-related functional impairment. In contrast, *hsa_circ_0003445* was significantly upregulated in RRMS with EDSS > 4.5 compared to RRMS with EDSS < 4.5, but significantly downregulated in SPMS with EDSS > 4.5. No significant difference was found in the expression levels of the other circRNAs analyzed (Figure 7A). We also examined the relative abundance of the cognate mRNAs for each circRNA and found a significant downregulation of *SAMD8* and *RPL36* mRNAs in RRMS with EDSS > 4.5, and *RPL36* mRNA in SPMS as compared to RRMS with EDSS < 4.5. Interestingly, the coherent downregulation of *hsa_circ_0018905* with MS functional impairment could drive alterations of the protein SMAD8 produced by the same parental gene. Additional analyses are needed to clarify this point. The RNAs transcribed from the parental genes (*HDAC4* mRNA and *RPPH1*) showed no significant difference (Appendix A).

To assess whether the differentially expressed circRNAs, which were differentially expressed in severe MS, could be candidate biomarkers for MS, we assessed the predictive value of *hsa_circ_0018905* and *hsa_circ_0003445* by employing the ROC curve analysis and using the Cq value for each circRNA. The results reported in Figure 7B showed that the area under the curve (AUC) is significant (*p* < 0.0002) for *hsa_circ_0003445,* with an AUC value of 0.83 (95% CI: 0.706–0.83), with a sensitivity of 90% and a specificity of 70%. For *hsa_circ_0018905*, we observed an AUC value of 0.948 (95% CI: 0.884–0.948) (*p* = 5.1 × 10^−8^), a sensitivity of 90%, and a specificity of 80%.

## 4. Discussion

Using circRNA microarray analysis, we discovered circRNAs differentially expressed in PBMCs of newly diagnosed MS patients as compared to HCs. The analysis identified 64 differentially expressed circRNAs, of which 11 circRNAs were upregulated and 53 were downregulated. Among the 64 differentially expressed circRNAs, 7 circRNAs were experimentally validated as dysregulated in the MS population using PBMCs from an independent validation cohort (*hsa_circ_0000518*, *hsa_circ_0000517*, *hsa_circ_0000514*, and *hsa_circ_0000511* were confirmed to be upregulated, whereas *hsa_circ_0018905*, *hsa_circ_0048764*, and *hsa_circ_003445* were confirmed to be downregulated). Due to their exceptional stability and high abundance in body fluids and EVs, circRNAs are considered promising biomarker candidates for non-invasive testing in clinical samples, including serum, plasma, saliva, urine, and cerebrospinal fluid. Interestingly, four of the validated circRNAs were also identified and validated in serum isolated from the blood of MS patients (*hsa_circ_0000518*, *hsa_circ_0000514*, *hsa_circ_0000511*, and *hsa_circ_0006853*). Notably, two circRNAs were confirmed to be downregulated in MS patients after all the validation steps, even when the disease course changed, as shown by a higher score using the EDSS and/or SP status; in particular, *hsa_circ_0018905* and *hsa_circ_003445* were observed to remain downregulated with worsening of the disease severity (from low to high EDSS and/or from RR-to-SP status). In this context, our finding is likely correlated with the different stages of the pathophysiological process. In most cases, the development of disability leading to reduced ambulatory capacity and consequent high EDSS is indeed associated with neurodegenerative processes. Conversely, the profile of patients with lower EDSS is typically characterized by a predominance of active focal inflammation.

All the upregulated circRNAs identified are transcribed from the *RPPH1* gene; *RPPH1* is the RNA component of the RNase P ribonucleoprotein, an endoribonuclease that participates in tRNA maturation [32]. Upregulation of the circRNAs generated from the *RPPH1* locus was validated in both PBMCs and serum (Figure 3 and Figure 4) but was not observed in cross-sectional data analyzing changes in disease status (Figure 7). The role of this locus, and in particular of the *hsa_circ_0000518* and *hsa_circ_0000517*, was previously described in MS [16]. Furthermore, *RPPH1* was recently identified as one of the top differentially expressed RNAs implicated in RNA processing in the B-cell transcriptome of MS patients [23]. In this context, alteration of the circRNAs expression encoded from the RPPH1 gene (e.g., *hsa_circ_0000518* and *hsa_circ_0000517)* described by our data may affect RPPH1 expression, thus influencing RNA processing in the B-cell and consequently B cell functions; indeed, circRNAs are well-known regulators of their linear parental gene through a preferential modulation of the parental gene transcription [33]. Of note, it has been observed by Jiang and colleagues that *hsa_circ_0000518* promotes macrophage/microglia M1 Polarization via the FUS/CaMKKβ/AMPK pathway to aggravate MS [34]. Further studies analyzing circRNA expression using a single-cells approach are needed to clarify the expression and role of each identified circRNA in the context of MS disease.

In general, *RPPH1* has been [35,36] involved in the inflammatory pathogenesis of diabetic nephropathy and in the proliferation of breast cancer cells. It has also been reported to have a role in the pathogenesis of Alzheimer’s disease (AD), where upregulation of *RPPH1* in AD mice was linked to increased expression of CDC42, promoting the formation of dendritic spines in hippocampal neurons [37], whereas *RPPH1* overexpression increased cell viability and inhibited apoptosis in Aβ-induced SK-N-SH cells [38]. While different studies suggest the role of *RPPH1* in the onset of different diseases, including MS, the exact function and mechanism underlying the role of *RPPH1* remain to be elucidated.

The validated circRNAs showing reduced abundance in MS were transcribed from different host genes—*SAMD8*, *RPL36*, and *HDAC4*. The mRNA encoding SAMD8 (Sphingomyelin synthase-related protein 1), and *hsa_circ_00018905* were both downregulated in PBMCs of MS patients, and they remained downregulated despite changes in disease status (Figure 7 and Appendix A) Furthermore, ROC analysis identified *hsa_circ_00018905* as a new promising candidate biomarker for MS. SAMD8 is an ER-resident ceramide phosphoethanolamine (CPE) synthase and a suppressor of ceramide-mediated apoptosis in cultured cells; it was predicted to enable ceramide cholinephosphotransferase activity as well as sphingomyelin synthase activity, and it is involved in ceramide biosynthetic and regulation of ceramide biosynthetic processes [39,40]. Sphingomyelin is present in all cell membranes and also regulates inflammatory signaling and immune cell function; therefore, it may have a role in the pathophysiology of inflammatory disorders, including MS [41,42]. SAMD8 has been already associated with neurodegenerative conditions like Huntington’s and AD while elevated SAMD8 expression in the brain raises questions regarding its possible role in neurodegenerative disorders that still need to be properly elucidated [40]. Regarding *hsa_circ_00018905,* no data are available today, and additional studies are necessary to elucidate how dysregulation is induced in MS samples and how it affects disease progression.

The RPL36 (Ribosomal Protein L36) gene from which *hsa_circ_0048764* is transcribed was mainly studied in glioma pathogenesis. It has been reported that downregulation of *RPL36* mRNA can inhibit cell proliferation and induce cell cycle arrest in glioma through STAT1. Analyzing expression data of common genes, Macrophage Migration Inhibitory Factor signaling dysregulation of the RPL36 has been found to be associated with MS [43,44]. However, the molecular and pathological connections with the disease have not been identified until now. Additionally, *hsa_circ_0048764* studies are still in their infancy and no connections have been identified with MS. Finally, for the transcripts generated by the *HDAC4* gene, we observed a downregulation in PBMCs for both *HDAC4* mRNA and *hsa_circ_003445*. Interestingly, the downregulation is maintained in the SP form of MS, but is not present when analyzing samples purified from patients with RRMS and higher EDSS. *Hsa_circ_003445* is also indicated as a possible biomarker for MS by ROC curve analysis (Figure 7A,B). HDAC4 encodes the histone deacetylase 4, a protein involved in histone modifications and highly abundant in the brain [45]. HDAC4 may play a role in neuronal plasticity, and there is some evidence that HDAC inhibitors lessen the neuropathy in mice used as MS models (EAE) [46]. Furthermore, emerging evidence suggests epigenetic modifications have a role in affecting the risk of MS [47]. While studying differentially methylated regions in case/control cohorts focusing on progressive MS, Maltby and colleagues found that genes *HTR2A*, *SLC17A9*, and *HDAC4* were differentially methylated between MS and control individuals [48]. Additionally, RNA sequencing analysis from different brain regions, at different stages of differentiation, uncovered thousands of highly abundant circRNAs in neurons, including *hsa_circ_0003445* [49]. To sum up, we confirm the dysregulation of circRNAs that are generated from the RPPH1 gene in MS and identify new circRNAs dysregulated with the disease. Of note, all of them are generated from genes that have been implicated, at different levels, with the regulation of immune functions, one of the critical components of MS. Considering the functions of circRNA, one can expect that the dysregulation of the indicated circRNAs may affect the functions of the linear parental gene, altering immune regulation.

CircRNAs are emerging as important regulators of the immune system, significantly influencing immune cell development and differentiation, which are strongly linked to the onset and progression of autoimmune diseases [50]. Functionally, circRNAs can influence the activation and activity of various immune cells (e.g., T cells, B cells, and macrophages), thereby fine-tuning immune responses [51]. CircRNAs are also involved in the regulation of cytokines production and essential signaling pathways related to immune responses [52]. Investigating the expression profiles of circRNAomes in B cells, T cells, and monocytes, Nicolet et al. identified significant differences in circRNA expression among different immune cells. Interestingly, this study highlighted hundreds of circRNAs that exhibit cell type-specific expression suggesting the involvement in the regulation or maintenance of specific cell functions [53]. Additionally, during hematopoietic differentiation, lymphocytes displayed the highest circRNA expression levels, reflecting a greater abundance rather than diversity. For instance, circ-FNDC3B showed the highest expression in natural killer cells, while circ-ELK4, circ-MYBL1, and circ-SLFN12L were predominantly expressed in T cells and natural killer cells. Moreover, Maass et al. examined circRNA expression profiles across 20 human tissues closely associated with various diseases. They demonstrated that many circRNAs exhibit tissue-specific expression, potentially linked to the clinical phenotypes and underlying mechanisms of human diseases [54].

Concerning the circRNAs differentially expressed in MS and identified in this report, no studies have analyzed the expression pattern in the immune cells population except for *hsa_circ_0000518*, as mentioned above, which has been deeply studied in macrophages in the context of MS. In their manuscript, Zhang et al. analyzed circRNA expression in macrophages under M1 (interferon-γ and LPS-induced) and M2 (IL4-induced) polarization conditions. *Hsa_circ_0000518* has been observed to be upregulated in the cerebrospinal fluid and peripheral blood of MS patients. Additionally, interfering with *hsa_circ_0000518* expression in vitro led to reduced FUS expression, promoted the polarization of LPS-activated HMC3 cells towards the M2 type, and alleviated CNS injury in an experimental autoimmune encephalomyelitis (EAE) mouse model, indicating that *hsa_circ_0000518* might be a potential therapeutic target for MS [34]. CircRNAs exert their functions as miRNA and RNA binding sponges that affect cell regulation at various levels. This regulatory process is implicated in various vital biological functions, such as the differentiation and survival of brain cells and the modulation of various immune processes including B-cell function, a key element of the MS pathology. In this regard, our analysis (Figure 5 and Figure 6) predicted that *hsa_circ_0000518* and *hsa_circ_0000517* may sponge miR-326, a microRNA highly abundant in active MS lesions in comparison with inactive lesions or normal brain white matter which was found to be upregulated during relapse phases in RRMS [55]. Functionally, miRNA-326 promotes T helper 17 cell differentiation and phagocytosis of myelin, and has been shown to be the target for the therapeutic inhibition of disease in mice used to study autoimmune encephalomyelitis [55]. CircRNAs may also bind to and sequester RBPs, affecting the interactions with other RNAs, thus modulating translation and RNA stability [56,57]. Jiayi et al. found that the RBP HuR and miR-29a controlled the production of cystatin F, a crucial inhibitor of papain-like lysosomal cysteine proteinases that plays a pivotal role in demyelination and remyelination. Reduced levels of both HuR and cystatin F were observed in the core regions of MS plaques compared to the border zone, implying a potential connection between decreased HuR levels and *cystatin F* mRNA instability, leading to exacerbated demyelination in MS patients [58]. In this context, even if the copies per cell of *hsa_circ_0000518*, *hsa_circ_0000514*, and *hsa_circ_0000511* are not known, the upregulation of these circRNAs may lead to the sequestration of HuR, possibly at a localized level, in turn exacerbating the demyelination process.

Several studies have now demonstrated that non-coding RNAs, including circRNAs and miRNAs, have been implicated in the development of MS at different levels. Indeed, circRNAs can interact with miRNAs, influencing their specific mRNA targets’ levels and shaping the so-called competing endogenous RNAs (ceRNA) network. Thus, we also investigated the circRNA-miRNA-mRNA network for MS risk-associated genes identified by the International Multiple Sclerosis Genetics Consortium [6]. Although ceRNA analysis must be considered with caution, after considering the intracellular stoichiometry of the regulatory RNAs, the possibility that some MS-associated circRNAs could regulate MS pathogenesis deserves closer scrutiny. For example, we showed a deregulation of the immune system, including T cell proliferation and differentiation (e.g., STAT3, TXK, and CXCR4) and neurodegeneration (e.g., FOXP1 and PLEC). Alterations in circRNA-miRNA-mRNA network might provide microenvironmental changes that modulate MS progression, such as *hsa_circ_0000518* and *hsa_circ_0000517* (upregulated), as well as *hsa_circ_0048764* and *hsa_circ_0003445* (downregulated), which were predicted to bind the different miRNAs that regulate the same gene, PLEC. PLEC encodes the cytoskeleton plectin protein, which is present in numerous tissues, including nervous tissue, and is involved in maintaining tissue integrity and elasticity [59]. Furthermore, Orton et al. observed consistent differences in PLEC gene expression in MS patients with different disease severity, reporting upregulation of PLEC in severe MS cases [60].

Until now, few studies have explored the relationship between circRNAs and MS. Among these, only two studies share similarities with ours: those by [16] and [30] I conducted a microarray analysis on PBMCs from four RRMS patients compared to four HCs and identified 406 differentially expressed circRNAs, with the top two upregulated, *has_circ_0000518* and *hsa_circ_0000517*, being consistent with our findings [16]. On the other hand, Zurawska et al. performed a microarray analysis on PBMCs isolated from MS patients during the relapse and remission phases and compared them with HCs [30]. However, these previous studies did not employ newly diagnosed patients, and the differences among them could be explained by heterogeneity due to disease progression and specific therapeutic interventions used by patients. Furthermore, our study, as well as the other described above [16,30], is characterized by an important limitation: the discovery dataset is small. In our study, for example, five MS samples vs. five HCs were used. Further studies should be carried out to characterize the expression profile and function of circulating circRNAs in a more extended dataset.

Although there is growing evidence highlighting the significant involvement of circRNAs in regulating immune cell functions and the CNS [61,62,63], their specific role(s) in the development of MS remain largely unexplored, with only a limited number of studies investigating this aspect. Our study focused on the expression profiles of circRNAs in MS patients at the moment of diagnosis, before therapy, improving our understanding of the role of circRNAs during the first phases of MS pathogenesis. However, to date, no direct link through genetic association with MS has been reported for the identified genes (*RPPH1, SAMD8*, *RPL36*, and *HDAC4*), thus suggesting a role for the encoded circRNAs as biomarkers, rather than in the causal biology of disease. In sum, for the first time, we have studied the expression profile of more than 13,000 circRNAs in MS patients at the time of diagnosis, linking circRNAs to MS, and highlighting for the first time a possible role for *hsa_circ_0018905* as a biomarker for MS.

## Figures and Tables

**Figure 1 cells-13-01668-f001:**
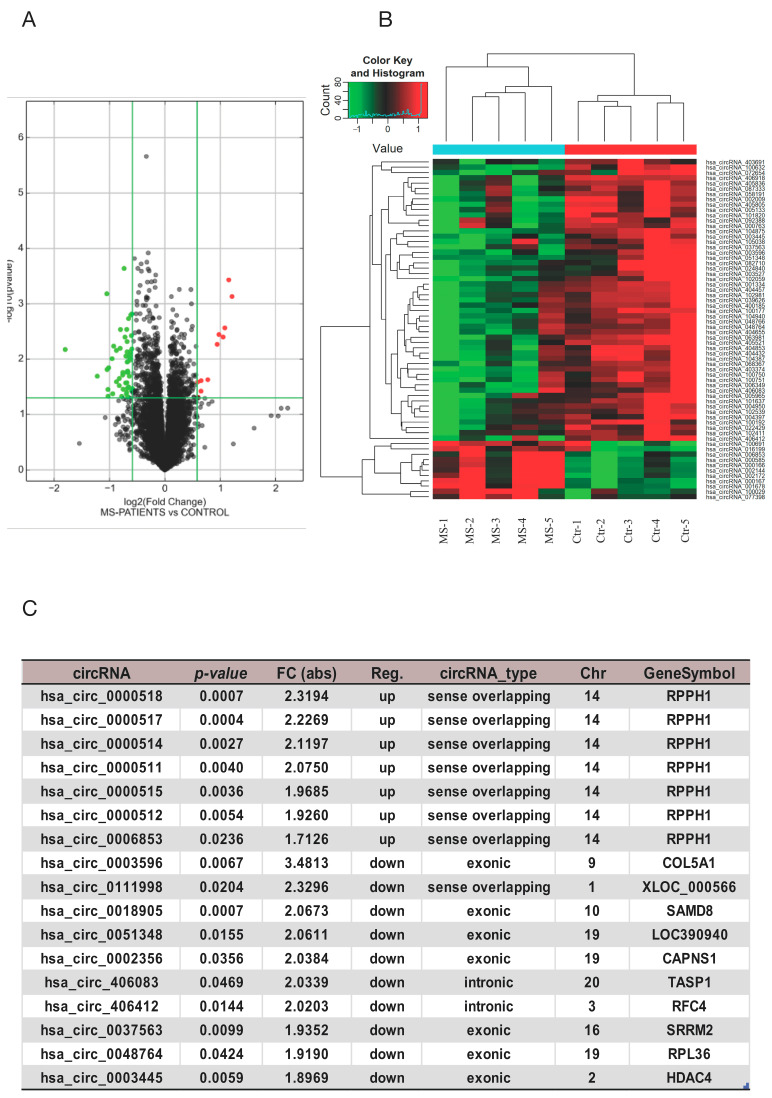
Differentially expressed circRNAs in MS patients versus HCs, circRNA array analysis. (**A**) Volcano plots, used to visualize up- and downregulated genes across MS samples as compared to HCs. The red (up) and green (down) dots in the plot represent the significative differentially expressed circRNAs. (**B**) Clustered heatmap of the differentially expressed circRNAs showing the relationships among the expression levels of samples. Upregulation is shown in red, and downregulation is in green. (**C**) Table showing the list of circRNAs differentially expressed, depicting the top 7 upregulated and 10 downregulated.

**Figure 2 cells-13-01668-f002:**
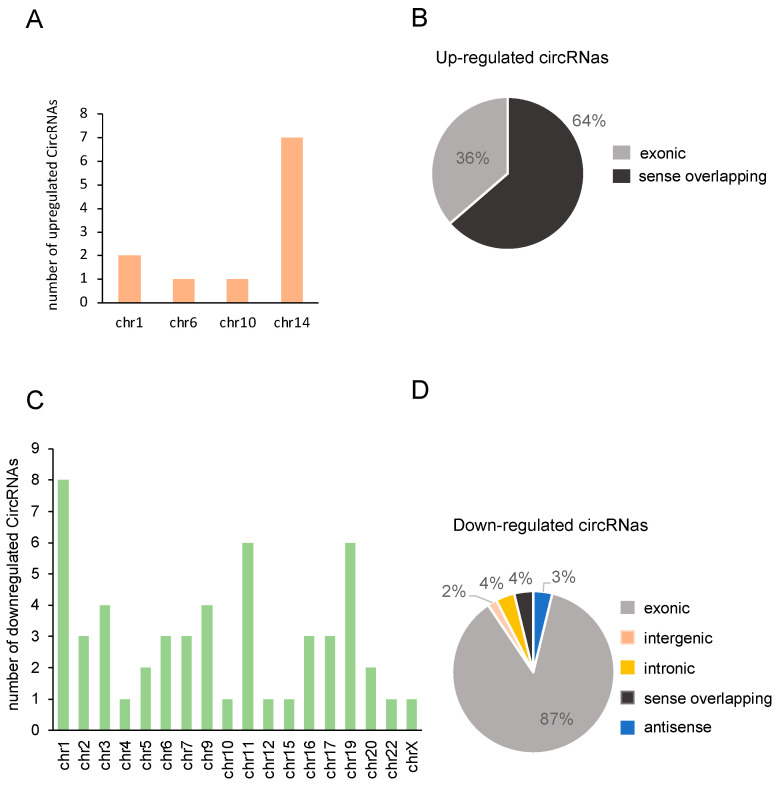
Characteristics of the circRNAs identified in PBMCs of MS patients versus HCs. (**A**) Distribution of significantly upregulated circRNAs according to the chromosomal location. (**B**) Class distribution of upregulated circRNAs based on the genomic origins. (**C**) Distribution of significantly downregulated circRNAs according to the chromosomal location. (**D**) Class distribution of downregulated circRNAs based on the genomic origins.

**Figure 3 cells-13-01668-f003:**
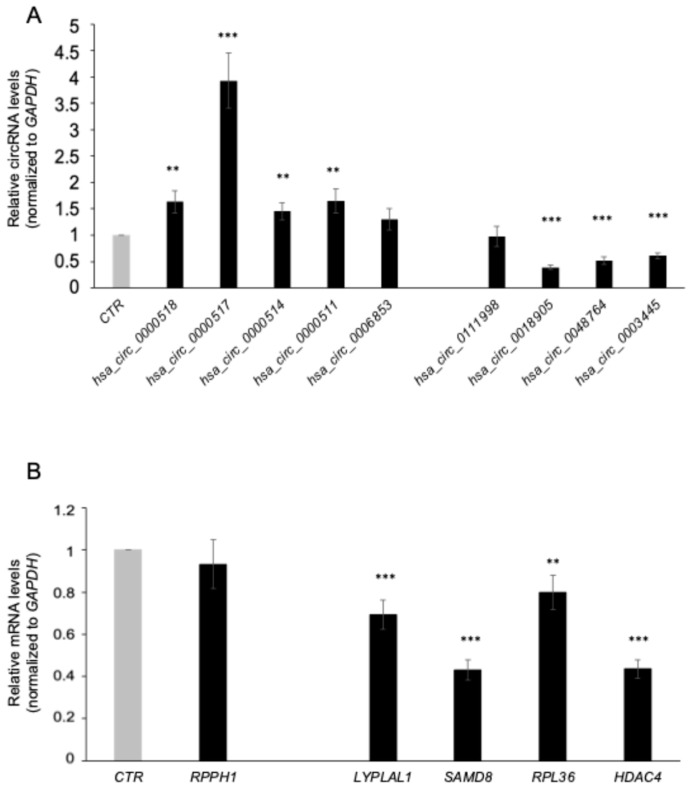
Validation of the circRNAs identified in PBMCs of MS patients versus HCs. Expression levels in PBMCs of five upregulated and four downregulated circRNAs (**A**) and the corresponding cognate linear mRNAs (**B**) were measured by qPCR analysis. The levels of circRNAs and mRNAs were normalized to *GAPDH mRNA* levels. Data are the means and standard deviation (+SD) from at least three independent experiments. ** *p* < 0.01, *** *p* < 0.001.

**Figure 4 cells-13-01668-f004:**
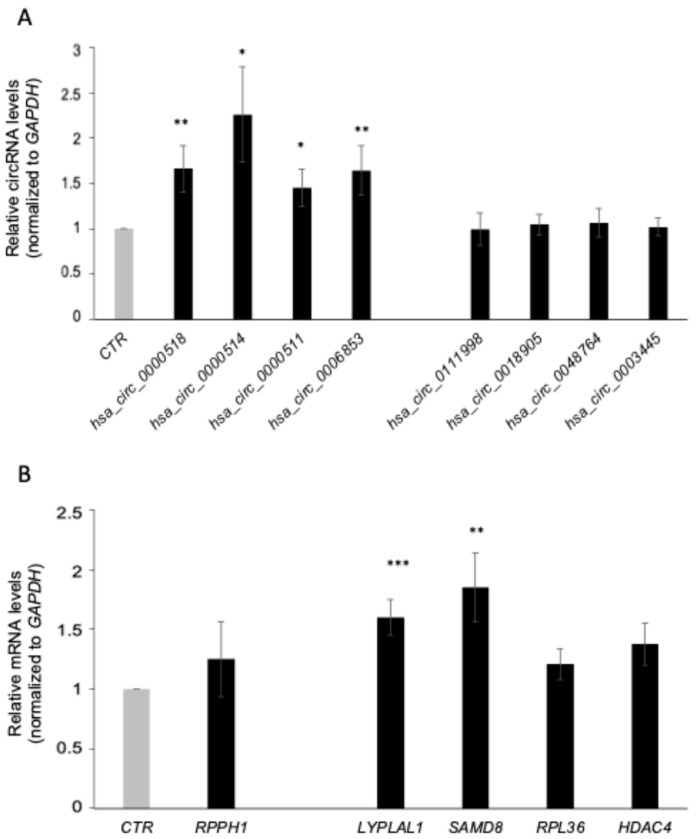
Validation of the circRNAs in serum of MS patients versus HCs. The levels in serum of five upregulated and four downregulated circRNAs (**A**) and the corresponding mRNAs (**B**) were measured by qPCR analysis. The levels of circRNAs and mRNAs were normalized to *GAPDH mRNA* levels. Data are the means and standard deviation (+SD) from at least three independent experiments. * *p* < 0.05, ** *p* < 0.01, *** *p* < 0.001.

**Figure 5 cells-13-01668-f005:**
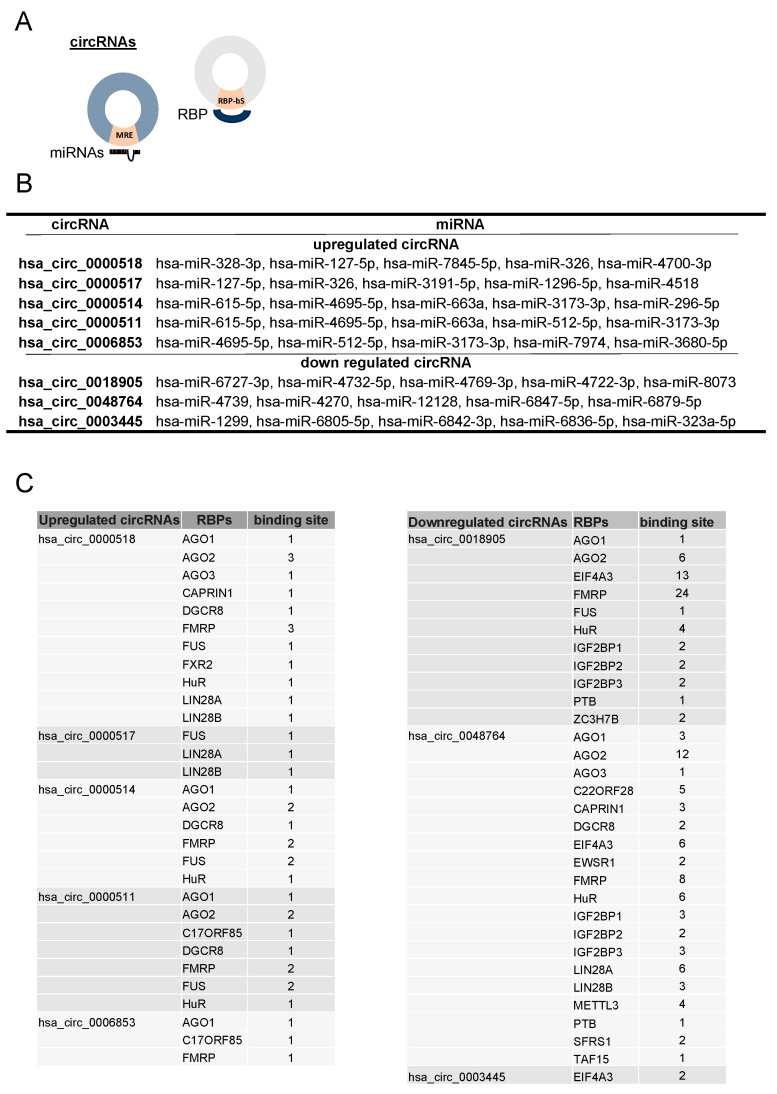
Identification of the miRNAs and RBP Targets. (**A**) Schematic representation of circRNAs with putative miRNA binding site (MRE) and RNA-binding protein binding site (RBP-bs). (**B**,**C**) Tables showing list of human circRNA identified from our studies and target miRNAs and interacting RNA-binding proteins as determined by analysis performed using miRanda and circInteractome, respectively.

**Figure 6 cells-13-01668-f006:**
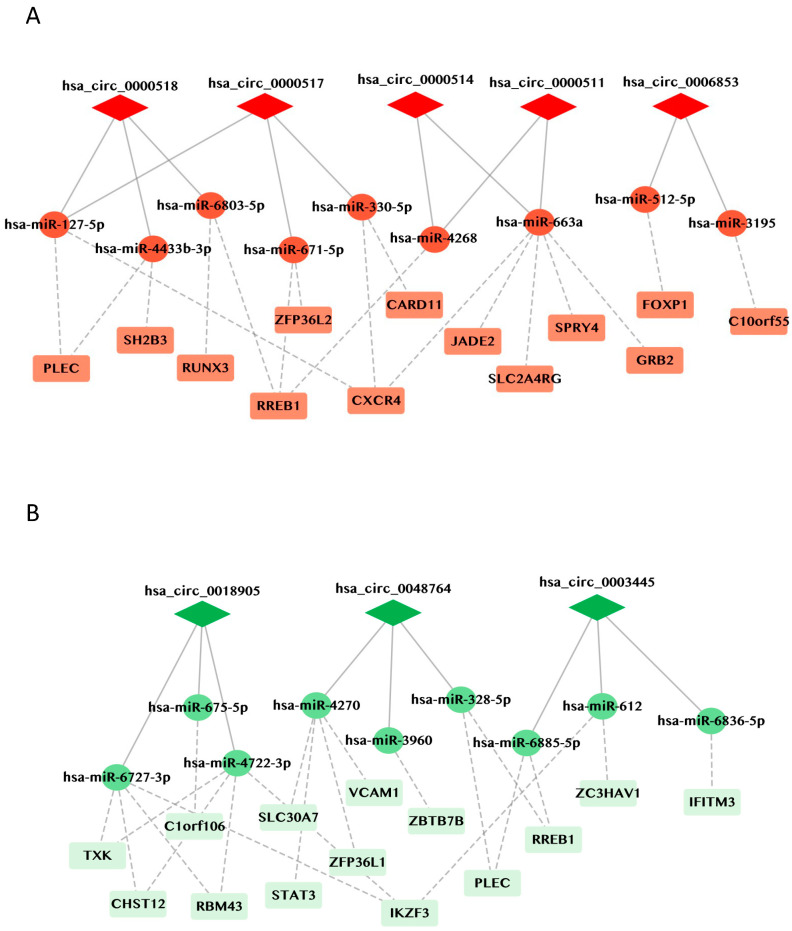
Network of circRNA-miRNA-mRNA for MS-associated genes. (**A**) Network of upregulated circRNAs and (**B**) downregulated circRNAs. CircRNAs are represented as red or green diamonds, miRNAs as red or green circles, and mRNAs as light red or light green rectangles. Red represents network generated from upregulated circRNAs and green from downregulated circRNAs.

**Figure 7 cells-13-01668-f007:**
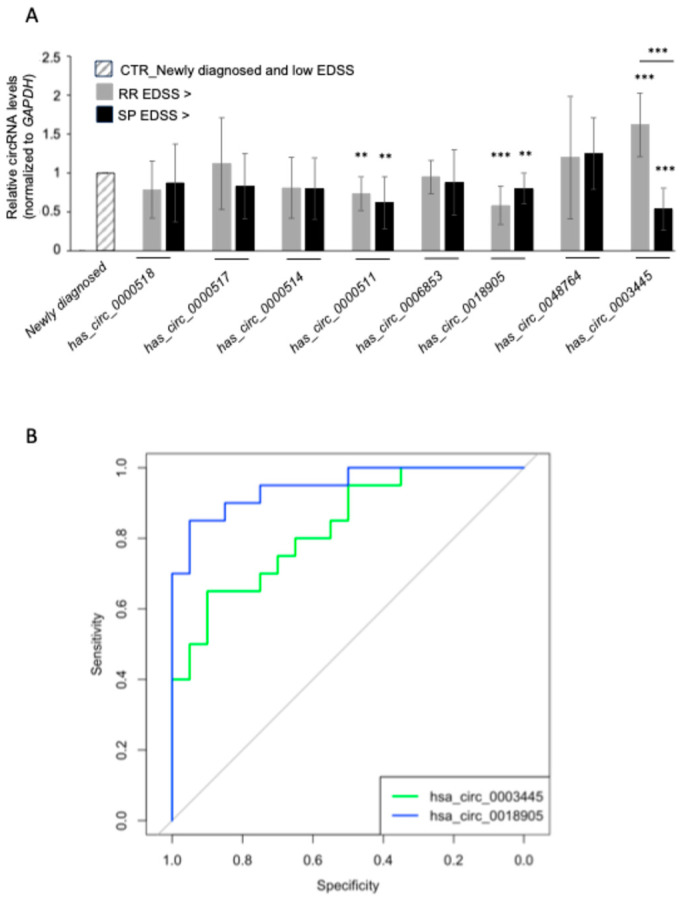
Validation of the circRNA expression in PBMCs and correlation with disease severity. *(***A**) Expression levels in PBMCs of five upregulated and three downregulated circRNAs in MS with different disease severity measured by RT-qPCR analysis. The levels of circRNAs were normalized to *GAPDH* mRNA levels. (**B**) Receiver operating characteristic (ROC) curve of differentially expressed circRNAs in MS vs. HCs. Green line, *hsa_circ_0003445* and blue line, hsa_circ_0018905. Data are represented as the means and standard deviation (+SD) from at least three independent experiments. ** *p* < 0.01, *** *p* < 0.001.

**Table 1 cells-13-01668-t001:** Gene-specific primer pairs.

RT-(q)PCR Primers	Primer Sequences (5′-3′)
** *hsa_circ_0000518* **	FW: CATGCCTACATTGCCCCAGA
	RV: CAGACCTTCCCAAGGGACAT
*hsa_circ_0000517*	FW: GGGAGGTGAGTTCCCAGAG
	RV: CAGGGAGAGCCCTGTTAGG
*hsa_circ_0000514*	FW: GAGCTTGGAACAGACTCACG
	RV: CATCTCCTGCCCAGTCTGA
*hsa_circ_0000511*	FW: CCTCCTTTGCCGGAGCTT
	RV: GGTCCACGGCATCTCCTG
*hsa_circ_0006853*	FW: TGAGTTCAATGGCTGAGGTG
	RV: GTTCCAAGCTCCGGCAAA
*hsa_circ_0111998*	FW: CAGCACTCCACAGCATCCACTA
	RV: TCCATTTCAATGGTAGCCTGCA
** *hsa_circ_0018905* **	FW: TGTTTGTGACCTCCCTCTCC
	RV: CTTCATCCTTCAGCCACACA
** *hsa_circ_0048764* **	FW: AAGCTGCTGCCAAGAAAGAC
	RV: ATCCTCCAACCTGCACAGAG
** *hsa_circ_0003445* **	FW: AGGAGCAGGAGCTGGAGAA
	RV: GTGCTGGGCATGTGGTTC
RPPH1	FW: CTGTCACTCCACTCCCATGT
	RV: TTCTCTGGGAACTCACCTCC
LYLPLAL1	FW: CTGCCCAGAACACCTTGAATC
	RV: TCCTGTTCTTCTTGATGCCAC
SAMD8	FW: TATGATCTCCGGTCTCCTCCT
	RV: TGTCACTGTTGTAGCCCATCT
RPL36	FW: GACCAAACACACCAAGTTCGT
	RV: TAAATTTGAGGGCCCGTTTGT
HDAC4	FW: AAAACGCAGCACAGTTCCC
	RV: GTCATCTTTGGCGTCGTACA
GAPDH	FW: ATTTGGTCGTATTGGGCGCC
	RV: TTGAGGTCAATGAAGGGGTC

**Table 2 cells-13-01668-t002:** Main characteristics of the individuals enrolled in the study.

**Discovery Set**	**HC**	**MS**
**No. tot**	5	5
**Female**	3	3
**Male**	2	2
**Age, yrs, mean ± SD**	34.8 ± 12.49	34.8 ± 12.49
**Disease duration, mean ± SD**	na	at diagnosis
**EDSS, mean ± SD**	na	2.3 ± 1.01
**Validation Set**	**HC**	**MS**
**No. tot**	20	20
**Female**	10	12
**Male**	10	8
**Age, yrs, mean ± SD**	48.75 ± 15.71	43.3 ± 14.23
**Disease duration, mean ± SD**	na	3.21 ± 3.89
**EDSS, mean ± SD**	na	2.5 ± 0.81
**MS Type**		RRMS

Abbreviations: SD, standard deviation; EDSS, Expanded Disability Status Scale; RRMS, relapsing–remitting MS; na, not applicable.

**Table 3 cells-13-01668-t003:** Main characteristics of the individuals enrolled in the second step of the validation process.

Disease Type	RRMS Edss < 4.5	RRMS Edss > 4.5	SPMS Edss > 4.5
**No. tot**	8	6	8
**Female**	6	5	4
**Male**	2	1	4
**Age, yrs, mean ± SD**	34.5 ± 11.95	64.3 ± 3.8	67.4 ± 7.2
**Disease duration, mean ± SD**	at diagnosis	25.8 ± 16.6	27.4 ± 12.1
**EDSS, mean ± SD**	2.3 ± 0.51	6 ± 0.70	6.3 ± 0.51

Abbreviations: SD, standard deviation; EDSS, Expanded Disability Status Scale; RRMS, relapsing–remitting MS; SPMS, secondary progressive MS.

## Data Availability

All data needed to evaluate the conclusions in the paper are present in the paper and in the Appendix A.

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
