# Peer review of "Identification of hsa_circ_0018905 as a New Potential Biomarker for Multiple Sclerosis"

_cells, 2024, doi:10.3390/cells13191668_

Round 1

Reviewer 1 Report

Comments and Suggestions for Authors

This manuscript investigates the role of circular RNAs (circRNAs) in Multiple Sclerosis (MS) by identifying hsa_circ_0018905 as a potential biomarker associated with disease severity. The study employs a comprehensive approach, including microarray and qRT-PCR analysis, to profile circRNA expression in peripheral blood mononuclear cells (PBMCs) from newly diagnosed MS patients and healthy controls. The authors present a novel circRNA-miRNA-mRNA network analysis, proposing regulatory roles for specific circRNAs in MS pathogenesis. This research contributes significantly to understanding non-coding RNA dysregulation in MS. It offers new potential biomarkers for disease progression, which could be crucial for early diagnosis and targeted therapeutic strategies. Overall, this work provides valuable insights and makes a noteworthy contribution to the field.

Detailed suggestions for improving the manuscript:

1.      The introduction could benefit from smoother transitions between concepts. Instead of jumping from general MS pathogenesis to circRNAs, include a bridge: for example: "While genetic and environmental factors play a crucial role in MS, growing evidence highlights the importance of RNA regulation, particularly non-coding RNAs, in disease development and progression……."

2.      In the introduction, more context on circRNAs’ molecular functions should be added beyond the mention of their closed-loop structure and stability.

3.      The cohort's exclusion of patients with comorbidities and treatment is mentioned but not fully detailed. Strengthen this by specifying what comorbidities were excluded and why, strengthening the study's conclusions. Also, clarify whether "no pharmacological treatment" refers to MS-specific therapies or any medications.

4.      The results section could expand the significance of hsa_circ_0018905's downregulation with disease severity.

5.      The figures are clear, but their descriptions in the text could be more integrated into the narrative. Instead of just stating that certain circRNAs were upregulated or downregulated, briefly discuss their potential roles.

6.      While the manuscript presents a solid dataset, it would benefit from a more explicit discussion of limitations, particularly the small sample size in the discovery cohort.

7.      The findings on serum vs PBMC circRNA levels are mentioned but not deeply explored. Discuss the implications of circRNA detection in serum, as this could enhance the biomarker's clinical utility for non-invasive testing.

8.      The circRNA-miRNA-mRNA network analysis is novel, but this point is not emphasized enough.

9.      The discussion touches on the therapeutic implications of targeting circRNAs like hsa_circ_0000518, particularly in immune regulation: "Given its role in macrophage polarization, targeting hsa_circ_0000518 could present new avenues for modulating immune responses in MS, especially during relapse phases."

10.   The discussion emphasizes upregulated circRNAs more heavily than downregulated ones. Since both play roles in disease, ensure equal attention is given to the biological significance of the downregulated circRNAs like hsa_circ_0018905 and their potential contribution to MS pathology.

Author Response

Comments: This manuscript investigates the role of circular RNAs (circRNAs) in Multiple Sclerosis (MS) by identifying hsa_circ_0018905 as a potential biomarker associated with disease severity. The study employs a comprehensive approach, including microarray and qRT-PCR analysis, to profile circRNA expression in peripheral blood mononuclear cells (PBMCs) from newly diagnosed MS patients and healthy controls. The authors present a novel circRNA-miRNA-mRNA network analysis, proposing regulatory roles for specific circRNAs in MS pathogenesis. This research contributes significantly to understanding non-coding RNA dysregulation in MS. It offers new potential biomarkers for disease progression, which could be crucial for early diagnosis and targeted therapeutic strategies. Overall, this work provides valuable insights and makes a noteworthy contribution to the field.

[AU] We thank the reviewer for his/her positive remarks.

Detailed suggestions for improving the manuscript:

  1. The introduction could benefit from smoother transitions between concepts. Instead of jumping from general MS pathogenesis to circRNAs, include a bridge: for example: "While genetic and environmental factors play a crucial role in MS, growing evidence highlights the importance of RNA regulation, particularly non-coding RNAs, in disease development and progression……."

[AU] We appreciate the reviewer’s helpful suggestion. We added few sentence to have smoother transitions between concepts in the introduction.

  1. In the introduction, more context on circRNAs’ molecular functions should be added beyond the mention of their closed-loop structure and stability.

[AU] We appreciate this concern and fully agree with the reviewer’s advice. We added more information on circRNAs function to the introduction; lines 70-74

  1. The cohort's exclusion of patients with comorbidities and treatment is mentioned but not fully detailed. Strengthen this by specifying what comorbidities were excluded and why, strengthening the study's conclusions. Also, clarify whether "no pharmacological treatment" refers to MS-specific therapies or any medications.

[AU] We thank the reviewer for asking that we clarify this point. Patients with any chronic comorbidity apart from MS were excluded to mitigate potential confounding variables in the analysis. Similarly, patients under drug treatments and with a history of recent infections were excluded from the study. The definition of 'no pharmacological treatment' indicates the absence of any medication. We believe that in this way the analyses can benefit from a more clear result were only the effect of MS are analyzed. (Lines 229-230)

  1. The results section could expand the significance of hsa_circ_0018905's downregulation with disease severity.

[AU] We appreciate the reviewer’s suggestions. Unfortunately, until now not a specific role have been associated with the down regulation of hsa_circ_0018905 in MS. Due to the general roles of cirRNAs in regulating cell biology we can hypothesize a possible change in the linear parental gene (SAMD8) that could drive downstream alterations. Lines 413-416

  1. The figures are clear, but their descriptions in the text could be more integrated into the narrative. Instead of just stating that certain circRNAs were upregulated or downregulated, briefly discuss their potential roles.

[AU] We thanks the reviewer for the observation, but the journal require to separate the results section from the discussion; thus, we provide a detailed description of each circRNA and linear counterpart are presented in the discussion section, lines 460 -531.

  1. While the manuscript presents a solid dataset, it would benefit from a more explicit discussion of limitations, particularly the small sample size in the discovery cohort.

[AU] We thank the reviewer for this observation; a sentence to point in explicit way limitations of the study have been added (Lines 619-623).

  1. The findings on serum vs PBMC circRNA levels are mentioned but not deeply explored. Discuss the implications of circRNA detection in serum, as this could enhance the biomarker's clinical utility for non-invasive testing.

[AU] Thank you for this comment. We added a sentence to better explain the relevance of detecting circRNAs in body fluids in the discussion section (Lines 445-448).

  1. The circRNA-miRNA-mRNA network analysis is novel, but this point is not emphasized enough.

[AU] We thank the reviewer for this observation. We pointed out the relevance of ceRNA network in the discussion section. (Lines 582-585)

  1. The discussion touches on the therapeutic implications of targeting circRNAs like hsa_circ_0000518, particularly in immune regulation: "Given its role in macrophage polarization, targeting hsa_circ_0000518 could present new avenues for modulating immune responses in MS, especially during relapse phases." 

[AU] Several lines of evidence point for a role of circRNAs in many cellular processes, with circRNA dysregulation related to the pathogenesis of various diseases including MS. CircRNAs present peculiar feature as compared to other class of RNA including the stability and a tissue- or cell type-specific expression. Consequently, several groups are now exploring the potential use of circRNAs as a therapeutic target. Of course, this approach has several limitations that can still need to be mitigated using innovative systems. In the manuscript we mentioned the possibility of using one of the dysregulated circRNA as drug target for MS. Of course, this field is still at his infancy, additionally it is not the focus of the manuscript, so we just mentioned this possibility regarding circRNAs.

  1. The discussion emphasizes upregulated circRNAs more heavily than downregulated ones. Since both play roles in disease, ensure equal attention is given to the biological significance of the downregulated circRNAs like hsa_circ_0018905 and their potential contribution to MS pathology.

[AU] We thank the reviewer for his/her concern. The paper describe and acknowledged the state of art literature for each circRNA identified by our analysis; upregulated circRNAs have an extended literature including already documented associations with MS. Regarding downregulated circRNAs like hsa_circ_0018905 and their potential contribution to MS pathology additional analysis are needed to clarify this point; thus, at the moment an equal attention cannot been guaranteed.

Reviewer 2 Report

Comments and Suggestions for Authors

Dear authors,

I understood reduction of circRNA associated with symptoms of MS and a circRNA might be identified as biomarker. This study is very impressive for me. However,  I think the manuscript was not suitable for publication in this journal.

Reviewer's comments as below.

1. In table 2, age of healthy controls and patients with MS is same. Is it correct? Please confirm it. 

2. Your table (table 2 and table 3) rows are not aligned properly. Please rearrange it.

3. I understand 5 people was newly diagnosed as MS. Others  did not receive any medical treatment for long time. Is it correct? How did authors know they are RRMS or another MS? 

Author Response

Dear authors,

I understood reduction of circRNA associated with symptoms of MS and a circRNA might be identified as biomarker. This study is very impressive for me. However,  I think the manuscript was not suitable for publication in this journal.

Reviewer's comments as below.

  1. In table 2, age of healthy controls and patients with MS is same. Is it correct? Please confirm it. 

[AU] Yes, it is correct. As described in the Materials and Methods section, we matched healthy controls and MS patients for both sex and age. Regarding age, we considered the year of birth, which is why the number is the same.

  1. Your table (table 2 and table 3) rows are not aligned properly. Please rearrange it.

[AU] Thank you for your comment. We have aligned the rows in the tables.

  1. I understand 5 people was newly diagnosed as MS. Others  did not receive any medical treatment for long time. Is it correct? How did authors know they are RRMS or another MS? 

[AU] Thank you for your comment. Yes, the reviewer is correct, the discovery cohort was composed of 5 MS patients and 5 matched HC that were not undertaking any medication. In the validation cohort was enlarged to have 20 MS patients and 20 HC. For MS diagnosis the recruiting neurologist collected data on disease course, classifyin patients according to available clinical data and on the basis of the patient's clinical course.

Round 2

Reviewer 1 Report

Comments and Suggestions for Authors

The authors have made commendable efforts to address most of the earlier concerns. I conclude with acceptance.